# How to GAN Higher Jet Resolution

Pierre Baldi[1], Lukas Blecher[2], Anja Butter[2], Julian Collado[1], Jessica N. Howard[3],
Fabian Keilbach[2*], Tilman Plehn[2], Gregor Kasieczka[4] and Daniel Whiteson[3]

**1** Department of Computer Science, University of California, Irvine, US
**2** Institut für Theoretische Physik, Universität Heidelberg, Germany
**3** Department of Physics and Astronomy, University of California, Irvine, US
**4** Institut für Experimentalphysik, Universität Hamburg, Germany

⋆ jnhoward@uci.edu

## Abstract

QCD-jets at the LHC are described by simple physics principles. We show how super-resolution generative networks can learn the underlying structures and use them to improve the resolution of jet images. We test this approach on massless QCD-jets and on fat top-jets and find that the network reproduces their main features even without training on pure samples. In addition, we show how a slim network architecture can be constructed once we have control of the full network performance.

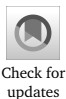 Check for updates

# 1 Introduction

Recent innovations in machine learning (ML) have provided boosts to many areas of particle physics. Ideas developed by the machine learning community to solve tasks unrelated to physics often have potential for applications within analysis of data in particle physics, even beyond improvements to analysis of high-dimensional data and speed improvements of first-principle simulations. One such recent development is the ability to enhance the resolution of images [1,2], by learning context-dependent general rules that can be applied to specific observations to generate estimates of higher-resolution versions of the observed images. Hadronic jets produced in collisions at the Large Hadron Collider (LHC) are obvious candidates for testing many ML-methods, as they are measured in large numbers, they come with a simple theoretical description, their complexity is balanced by their local detector patterns, and they are an integral part of almost every LHC analysis. In this paper, we apply super-resolution methods to LHC jets for the first time, generating images of jets at significantly higher resolution than the original observations.

The idea of using ML methods for exploring jets has a rich history. Early jet classification studies date to the early 1990s [3,4], and work has recently gained momentum through applications of deep learning tools to low-level jet observables organized as calorimeter images [5–10]. This approach can also be applied to the theoretically and experimentally well-defined task of top-quark tagging [11,12]. An alternative approach to organizing calorimeter deposits as pixelated images is to prepare a list of the 4-momenta of subjet constituents [13–16], including recurrent neural networks inspired by language recognition [17,18] or point clouds [19–23]. These various approaches have been compared in detail [24], revealing that their expected performance in tagging hadronically-decaying top quarks is relatively independent of the motivation and the architecture of the network. Open questions include attempts to gain theoretical understanding of the network's learned strategy [25–28], the stability with respect to detector effects [29,30], treatment of the uncertainty [31,32], extension to a wide range of inputs [20], and anomaly detection [22,33–35].

The first of these open questions inspires us to search for ways to apply machine learning to improve experimental jet measurements, by combining the basic rules of jet physics with the specific information of an observed jet. Independent of the nature of a given jet, its physics is described by relatively few ingredients, most notably collinear and soft QCD splittings, which can be measured at the LHC [36]. These basic principles can allow a super-resolution algorithm [1,2] to accurately estimate the higher-resolution information that led to the observed results. Super-resolution algorithms are widely used in image applications [37,38], including those which use convolutional neural networks CNNs [39]. They can be combined with generative networks [40,41], which can describe jets [42–47] and LHC events [48–51] and have the potential to increase the speed of LHC event generators significantly [52–56]. Such super-resolution GANs [57,58] have already been applied to cosmological simulations [59,60].

A simple super-resolution task in jet physics is to improve the resolution of a calorimeter image, using general QCD patterns [61]. It raises the question of *whether an up-sampled jet image can include more information than the original, low-resolution image*. Naively, it seems that the answer must be no, based on the same reasoning that motivates the argument that a generative network cannot produce more information than exists in its statistically limited training data set. However, this argument fails to account for the implicit knowledge embedded in the architecture of the network, which can contribute information in the same manner as a functional fit [62]. A super-resolution network applied to LHC jets combines the information from the low-resolution image with QCD knowledge extracted from the training data, for instance the underlying theoretical principles of soft and collinear splittings combined with mass drop patterns. While we will not attempt to quantify the added information (such an

answer will depend on individual applications), we will show that super-resolution networks can enhance calorimeter images, and that training on QCD-jets vs top-quark jets indicates that model uncertainties for this application are small. Our detailed study of super-resolution for jets follows similar studies in Ref. [61] of single particles, on the way to wider applications of super-resolution networks in particle physics. For example, such networks can automatically test the consistency of a data set when applied to different layers of a calorimeter. With an appropriate conditioning, they can become elements of a tagging algorithm. Up-sampling from calorimeter to tracker resolution can provide consistency tests between charged and neutral aspects of an event and can be turned into a new way of identifying and removing pile-up. This is especially promising, as both sides of the up-sampling (low-resolution calorimeter data and high-resolution tracking data) are present and thereby allow training from data only.

## 2  Super-resolution GAN for Jets

**Jet images**    The task for our super-resolution networks is to generate a high-resolution (HR), super-resolved (SR) version of a given low-resolution (LR) image. While it is ill-posed in a deterministic sense, as many distinct HR images can correspond to a single LR image, it is well-defined in a statistical sense.

To test and benchmark our new method we use the standard, public top-tagging data set [14,33], which is also the basis of the community paper on supervised top-tagging [24]. It contains jet images from $t\bar{t}$ and QCD di-jet events generated with PYTHIA [63] for a center-of-mass energy of $\sqrt{s} = 14$ TeV, with DELPHES [64] used to model the ATLAS detector response, and with clustering and jet-finding done with FASTJET [65]. The fat anti-$k_\mathrm{T}$ jets [66] have a radius $R = 0.8$ and

$$p_{\mathrm{T},j} = 550 \dots 650 \text{ GeV} \qquad \text{and} \qquad |\eta_j| < 2 \,, \tag{1}$$

to have access to decent experimental resolution. The jet images are defined by pixel-wise $p_\mathrm{T}$, with order of 50 active pixels. This means that, for instance, images with $160 \times 160$ pixels have a sparsity of 99.8%. For the training of super-resolution models, we provide paired LR/HR jet images, which are generated by down-sampling the HR image. We use sum pooling on the jet constituents as an approximation to reduced detector resolution before we perform jet finding [67]. After jet finding, we select the hardest jet in each of the HR and LR images as a candidate pair, rejecting the pair if either jet has fewer than 15 constituents. To ensure that the selected HR-clustered and LR-clustered jets correspond to the same hard parton, we require the angular distance between the two to be $\Delta R = \sqrt{\Delta\eta^2 + \Delta\phi^2} < 0.1$. This procedure defines paired HR and LR jet images, where the LR jet image contains no information from the HR image. We apply this procedure to create LR-HR image pairs with down-scaling factors of 2, 4, and 8, removing events that fail the requirement for any particular resolution from all samples, which ensures that all jet samples contain the same set of events.

There are multiple ways of normalizing jet images to be better suited for machine learning. Such transformations do not retain the absolute momentum, which may not be a problem for classification, but for our purposes this information is needed. In Fig. 1, we show typical energy distributions after re-scaling the pixel entries with a power $p$. Clearly, some kind of re-scaling is helpful to enhance the otherwise extremely peaked spectrum. On the other hand, we know that the low-energy radiation is largely noise, which means that choosing $p$ too small is not helpful for the network to learn the leading patterns. We find that $p = 0.3$ is a good compromise, to be combined with the original image $p = 1$.

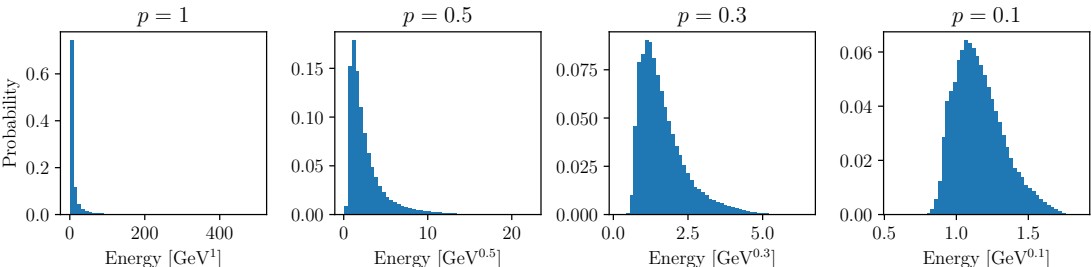

Figure 1: Distribution of energy deposition when pixel entries are raised by several different powers $E \rightarrow E^p$.

**Network architecture**   In our jet image study, we use a variant of the enhanced super-resolution GAN (ESRGAN) [58], illustrated in Fig. 2. To begin with, the generator converts a LR image into a SR image using a deep residual fully convolutional network. Its main element is the dense residual block (DRB) [68], built out of consecutive convolutional layers with $(3 \times 3)$-kernels, stride 1, padding 1, and 64 filters. The activation function is a LeakyReLU with $\alpha = 0.2$. The particularity of the DRB is that a layer receives the input of all other layers in addition to the output of the previous layer. This structure fuses all the feature maps inside the block. Three DRBs form a residual-in-residual dense block (RRDB) [58], connected via residual connections.

All convolutions in the generator preserve the spatial dimensions of the input image. Following Fig. 2, the up-sampling can be done by pixel-shuffle layers [69] or transposed convolutions. Our generator up-samples by a factor of two in up to three consecutive steps and works best if we alternate between pixel-shuffle and transposed convolutions. In the HR feature space, there are two additional convolutional layers, one of which simply scales the output by a fixed value.

The discriminator network is a relatively simple feed-forward convolutional network with LeakyReLU activations, as proposed for the SRGAN [57]. It uses blocks consisting of two convolutional layers with a $(3 \times 3)$-kernel and padding 1. While the first convolution of each block conserves the spatial dimensions, the second layer halves it through a strided convolution. We link four of those blocks and start with 64 filters, doubling the number of filters after each block. We modify the original SRGAN structure by removing the batch normalization layers and adding a gradient penalty [70–72]. We cut off the network before flattening, feeding it into a fully connected layer and switching to a Markovian discriminator. Finally, we include a second discriminator with exactly the same structure, such that the full discriminator response is the sum of two discriminator networks. For the second discriminator, we reset all weights after a fixed number of batches.

**Loss function**   The SRGAN and ESRGANs include a set of excess functionalities, such as perceptual loss which can potentially improve the quality of the output. This loss combines

Table 1: Sets of hyperparameters used for networks described in Fig. 2. Two sets are presented, one which optimized performance, and a second which which performed slightly worse. $\beta$ is the residual scale factor.

|  | #RRDB | batch size | $\beta$ | rescaling | $\lambda_{\text{reg}}$ | $\lambda_{\text{std}}$ | $\lambda_{\text{pow}}$ | $\lambda_{\text{HR}}$ | $\lambda_{\text{LR}}$ | $\lambda_{\text{adv}}$ | $\lambda_{\text{patch}}$ | reset interval |
|---|---|---|---|---|---|---|---|---|---|---|---|---|
| optimal | 10 | 15 | 0.1 | 0.3 | 0.001 | 0.2 | 1 | 1 | 0.1 | 0.01 | 0.1 | 20k |
| medium | 15 | 15 | 0.1 | 0.3 | 0.001 | 1.2 | 1 | 1 | 0.1 | 0.05 | 0.1 | 20k |

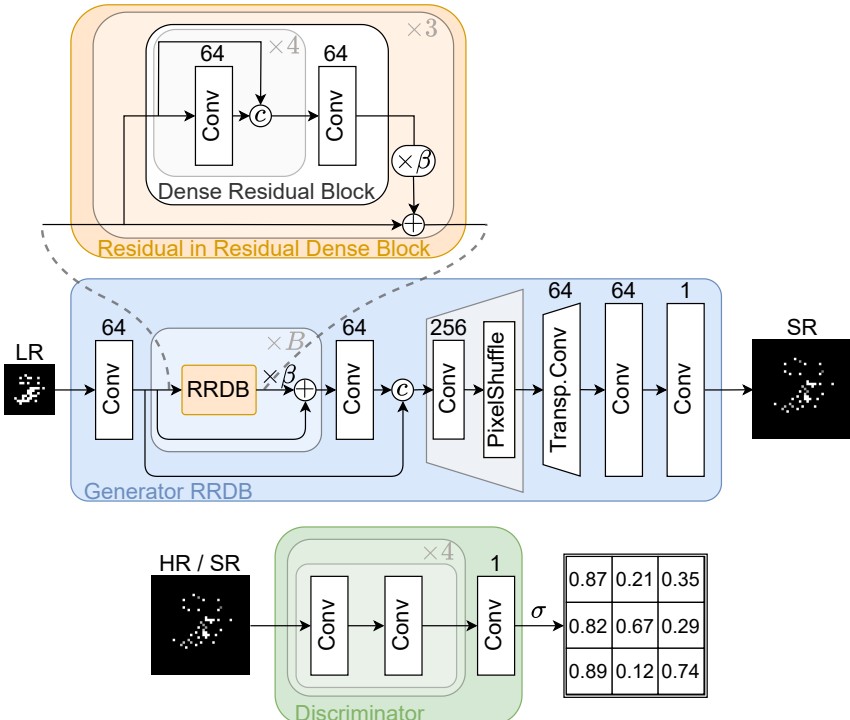

Figure 2: Architecture of modified generator network from ESRGAN (upper) and discriminator network modified from the SRGAN (lower).

the adversarial loss from the discriminator with a content loss that compares feature maps of a pre-trained image classification network. The adversarial loss for a relativistic GAN trained on true events ($T$) to generate new events ($G$) is

$$
\begin{aligned}
L_{\mathrm{adv}} &= -\langle \log D \rangle_G - \langle \log(1-D) \rangle_T \,, \\
\text{with} \quad D_T &= \sigma \left( C_T - \langle C \rangle_G \right) , \\
D_G &= \sigma \left( C_G - \langle C \rangle_T \right) ,
\end{aligned}
\tag{2}
$$

where $\sigma$ is a sigmoid classifier function and $C$ is the unactivated discriminator output. Compared to a standard adversarial loss, we have an additional term because $D_{\mathrm{T}}$ depends on the generated data $G$. The original content loss is not needed for our purpose. Because our HR images should resemble the ground truth, we add a $L_1$ loss between the SR and HR images. Our choice of $L_1$ over $L_2$ prevents blurring,

$$
L_{\mathrm{HR}} = L_1 (\mathrm{SR}, \mathrm{HR}) \,.
\tag{3}
$$

In return, because the LR image should correspond to the HR-jet, we define a loss term that compares the model input with the down-sampled model output pixel by pixel,

$$
L_{\mathrm{LR}} = L_1 \left( \sum_{\mathrm{pool}} (\mathrm{SR}), \mathrm{LR} \right) .
\tag{4}
$$

When we up-sample the LR-jet image by a factor $f$, we need to distribute each LR pixel energy over $f \times f$ SR pixels. These $f \times f$ pixels define a patch, and we encourage the network to spread the LR pixel energy such that the number of active pixels corresponds to the HR truth. This defines the loss term

$$
L_{\mathrm{patch}} = L_2 (\mathrm{patch}(\mathrm{SR}), \mathrm{patch}(\mathrm{HR})) \,.
\tag{5}
$$

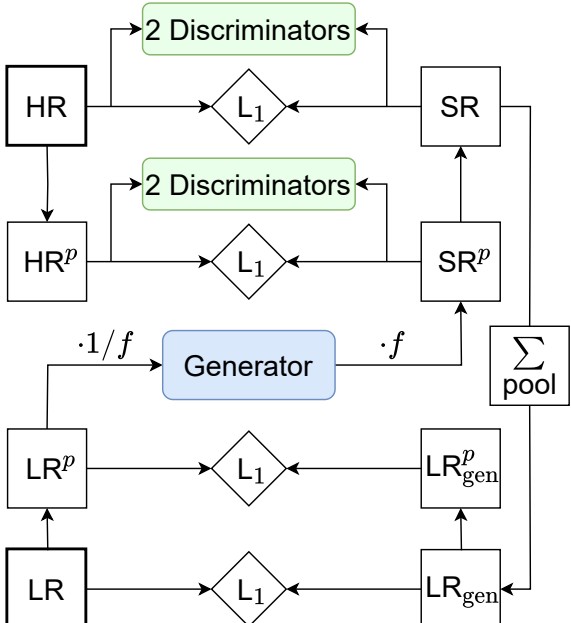

Figure 3: Training process for jet images. The generator and discriminator networks are shown in Fig. 2.

This patch loss is strictly not necessary for the network, but with the right choice of hyperparameters it balances the spread of constituents in the jet image plane with the appearance of artefacts. The combined generator loss over the standard and re-weighted jet images is then

$$L_G = \sum_{s \in \{\text{std, pow}\}} \lambda_s \left( \lambda_{\text{HR}} L_{\text{HR}} + \lambda_{\text{LR}} L_{\text{LR}} + \lambda_{\text{adv}} L_{\text{adv}} + \lambda_{\text{patch}} L_{\text{patch}} \right) , \qquad (6)$$

Our analysis included a careful scan over the hyperparameters $\lambda_j$. The metric we used to evaluate the performance of the models is the KL-divergence between the observables shown for instance in Fig. 4, most notably the energy of the hardest constituent and the average patches.

The GAN discriminator $D$ measures how close the generated data set $G$ is to the true or training data $T$. In a relativistic average GAN [72], the discriminator is given by the probability of a generated event being more realistic than the average true event, and vice versa. It corresponds to the adversarial generator loss in Eq.(2) but with switched labels,

$$= -\langle \log(1-D) \rangle_G - \langle \log D \rangle_T . \qquad (7)$$

To this expression we add a gradient penalty for stabilization,

$$L_{\text{reg}} = \langle (\| \nabla_{X'} C(X') \|_2 - 1)^2 \rangle , \qquad (8)$$

where $X'$ is a randomly weighted average between a real and generated samples, $X' = \epsilon X_T + (1-\epsilon) X_G$ and C(X') is the unactivated discriminator output.

All hyperparameters are listed in Tab. 1. We use ADAM [73] for the optimization with $\beta_1 = 0.5$ [74] and $\beta_2 = 0.9$. The learning rate is $\lambda = 0.0001$. The training of a model typically takes 50k-100k iterations.

**Training** The starting point of our training, illustrated in Fig. 3, is the HR truth image, from which the LR image is derived. All jet images are also raised to the power $p = 0.3$ as a pixelwise operation. We work with an up-scale factor $f = 2^3 = 8$. In that case, we divide the LR

image by the total factor $f$ and feed it into the RRDB generator. Its output is divided by the factor $1/f$ and gives the SR image raised to the power $p$. This intermediate result is saved for the computation of $L_{HR}$. For the SR output image we need to only take the $p^{th}$ root. This SR image is sum-pooled back to its LR version $LR_{gen}$ to compute the different generator loss terms. Based on this set of LR, HR, and SR images, with and without a $p$-scaling, we compute a set of $L_1$ loss contributions to the generator loss, as well as the discriminator losses from the HR-SR comparison.

# 3 Up-sampling jets

We benchmark the performance of the super-resolution algorithm for both QCD jets and top-quark jets. QCD jets, which at the LHC arise from massless partons, exist in large samples and are well described by collinear and soft splittings. As an alternative, we use jets from top-quark decays, which are significantly different, but can be isolated experimentally from semi-leptonic top-quark pair production and well-described theoretically via perturbative QCD.

We start with a set of HR-jet images with $160 \times 160$ pixels. We down-sample each of these images to a corresponding LR image by a linear factor $1/f = 1/8$ to an image of $20 \times 20$ pixels. For the up-sampling, we apply three doubling steps using pixel shuffle, transposed convolution, and another pixel shuffle. The pixel shuffle has the advantage of encoding the full information from the feature maps. It simply redistributes the information by transforming a large number of channels, as usually arise after deep convolutions, into a set of feature maps with fewer channels but larger spatial dimensions. The transposed convolution takes into account local information through a trainable kernel. After learning meaningful weights it can help learning intricate, non-local patterns, which would be missed by a global pixel shuffle. In the following, we first train and test a network on QCD-jets, then on top-jets. To estimate the model uncertainties, we apply networks trained on one class to the other class.

To evaluate the quality of the information in our image-based results in a physics context, we calculate an established set of jet observables [29, 75–77]

$$m_{jet} = \left( \sum_i p_i^\mu \right)^2, \qquad w_{pf} = \frac{\sum_i p_{T,i} \Delta R_{i,jet}}{\sum_i p_{T,i}},$$
$$C_{0.2} = \frac{\sum_{i,j} p_{T,i} p_{T,j} (\Delta R_{i,j})^{0.2}}{(\sum_i p_{T,i})^2}, \qquad \tau_N = \frac{\sum_k p_{T,k} \min(\Delta R_{1,k}, ..., \Delta R_{N,k})}{\sum_k p_{T,k} R_0}. \qquad (9)$$

The jet mass is the most relevant difference between pure QCD jets and top decay jets. The girth $w_{pf}$ essentially describes the geometric extension of the hard pixels, while $C_{0.2}$ is the leading pixel-to-pixel correlation. The subjettiness ratios $\tau_2/\tau_1$ and $\tau_3/\tau_2$ can distinguish between 2-prong and 3-prong decay jets.

## 3.1 Performance in QCD Jets

In an initial test, we train and test our super-resolution network on the sample of QCD jets, which are characterized by a few central pixels which carry most of the jet energy. In this case, it is important to include down-sampled kinematic distributions in the evaluation, to disentangle the central patterns.

In Fig. 4 we compare the HR and SR images as well as the true LR image with their generated $LR_{gen}$ counterpart. In addition to average SR and LR images, we show the energy spectra for the leading four pixels. This reveals how the LR image resolution reaches its limits, because the leading pixel carries most of the information. The sub-leading pixels are often harder for



**Figure 4:** Demonstration of the performance of a network trained on QCD-jets and applied to QCD-jets. Top left are averages of the HR and SR images, followed by distributions of the square-root of the energy of leading pixels, sub-leading, etc. Also shown are average $(f \times f)$-patches for the SR and the HR images, and distributions of high-level jet observables, see text for definitions. The zero-bin in energy collects jets with too few entries.

the HR image, because the up-sampling often splits the hardest LR pixel. From the 7th leading pixel and beyond, we see an increasing number of empty pixels, and above the 10th pixel the QCD jet largely features soft noise. This transition is the weak spot of the SR network. While it learns the underlying principles of QCD splittings for the hard pixels and the noise patterns for the soft pixels, the mixed range around the 7th and 10th pixels indicates sizeable deviations. We also show the average $(f \times f)$-patches for the SR and the HR images to confirm that the spreading of the hard pixels works at the 20% level.

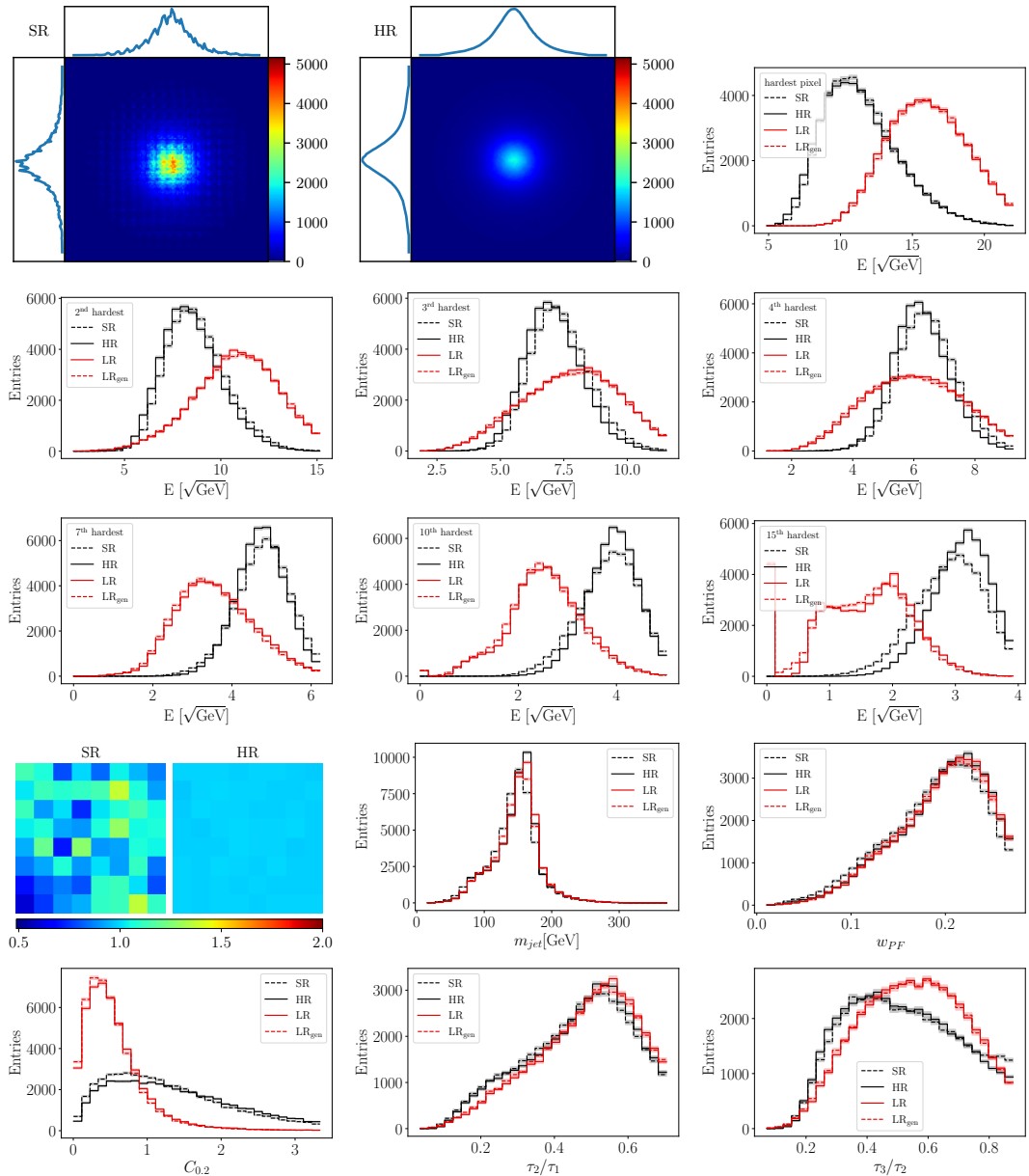

Figure 5: Demonstration of the performance of a network trained on top-quark jets and applied to top-quark jets. Top left are averages of the HR and SR images, followed by distributions of the square-root of the energy of leading pixels, sub-leading, etc. Also shown are average $(f \times f)$-patches for the SR and the HR images, and distributions of high-level jet observables, see text for definitions. The zero-bin in energy collects jets with too few entries.

Again in Fig. 4 we see that the jet mass peaks around the expected 50 GeV, for the LR and for the HR-jet alike. Still, the agreement between LR and LR$_{\text{gen}}$ on the one hand and between HR and SR on the other is better than the agreement between the LR and HR images. A similar picture emerges for the $p_{\text{T}}$-weighted distance to the jet axis, the girth $w_{\text{pf}}$, which essentially describes the extension of the hard pixels. The pixel-to-pixel correlation $C_{0.2}$ also shows little deviation between HR and SR on the one hand and LR and LR$_{\text{gen}}$ on the other. Finally, we see how the specific subjettiness ratios $\tau_2/\tau_1$ and $\tau_3/\tau_2$ increase for the HR/SR images, because

the splitting of hard central pixels into two hard and collinear, now resolved pixels increases the IR-safe subjet count. The ratio $\tau_3/\tau_2$ turns out to be one of the hardest of the HR-patterns to learn, with the effect that the SR version leads to slightly smaller values. This implies that the SR network does not generate quite enough splittings. Such a feature could of course be improved, but any optimization has to be balanced with the ability of the network to also describe jets with more than just collinear splittings, as we will see in the next case.

## 3.2 Performance in Top-Quark Jets

The physics of top-quark, light-quark, and QCD jets is very different. While for QCD-jets collinear and, to some degree, soft splittings describe the entire object, top-quark jets include the two electroweak decay steps. Comparing the top-quark jets shown in Fig. 5 with the QCD jets in Fig. 4 we see this difference already from the jet images — the top-quark jets are much wider and their energy is distributed among more pixels. From a SR point of view, this simplifies the task, because the network can work with more LR-structures. Technically, the adversarial loss becomes more important, and we can indeed balance the performance on top-quark jets vs QCD jets using $\lambda_{\text{adv}}$.

Looking at the ordered constituents, the additional mass drop structure is learned by the networks extremely well. The leading four constituents typically cover the three hard decay sub-jets, and they are described even better than in the QCD case. Starting with the 4th constituent, the relative position of the LR and HR peaks changes towards a more QCD-like structure, so the network starts splitting one hard LR-constituent into hard HR-constituents. This is consistent with the top-quark jet consisting of three well-separated patterns, where the QCD jets only show this pattern for one leading constituent. We also see that up to the 15th constituent, the massive top-quark jet shows comparably distinctive patterns and only few empty pixels.

For the high-level observables, we first see that the SR network shifts the jet mass peak by about 10 GeV and does well on the girth $w_{\text{PF}}$, aided by the fact that the jet resolution has hardly any effect on the jet size. As for QCD-jets, $C_{0.2}$ is no challenge for the up-sampling. Unlike for QCD-jets, $\tau_3/\tau_2$ is as stable as $\tau_2/\tau_1$, because it is completely governed by the hard and geometrically well-separated hard decays.

## 3.3 High-level observable benchmarking

To understand the physics impact of our super-resolution approach and the possible gains, we inspect distributions of the benchmark high-level observables defined in Eq.(9) and the subjettiness ratios $\tau_{1,2,3}$. In a first step, we compare the one-step by a factor eight with a more continuous up-sampling. In Fig. 6 we see that the three different down-scaling steps indeed interpolate between the full HR and LR jets smoothly. While the maximum in the number of active pixels shifts almost linearly, the jet mass is stable and not affected much by down- and up-sampling at all. The $p_{\text{T}}$-weighted girth is only affected for the collimated QCD jets, similar to the subjettiness ratio $\tau_2/\tau_1$. In contrast, the correlator $C_{0.2}$ indicates a continuous loss in resolution for top-quark jets, as does the subjettiness ratio $\tau_3/\tau_2$.

The same high-level observables also allow us to tackle the key question, whether up-sampling adds information for an individual jet. The source of the additional information comes from trends in jet properties the network has learned for top-quark or QCD jets, rather than a magic recovery of lost resolution. We remind ourselves the generative network is an inherently stochastic process, which provides a single example from a set of possible higher-resolution jets consistent with the observed lower-resolution jet, not a guess as to the true high resolution jet. Even a perfect network would display significant differences between generated jets and the single true jet. Instead, we can evaluate the information added by comparing how

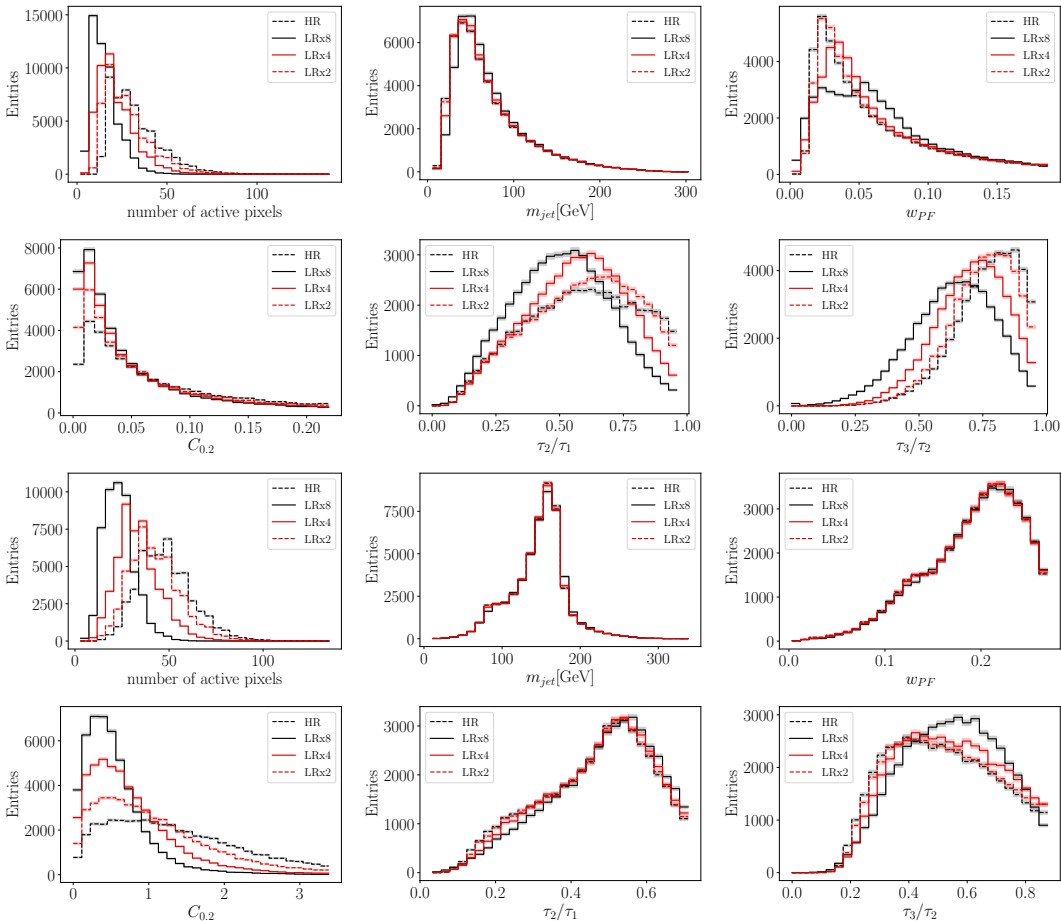

Figure 6: Distribution of the number of active pixels and high-level observables $m_{\text{jet}}$, $\tau_2/\tau_1$, $\tau_3/\tau_2$, $w_{\text{pf}}$, and for images down-scaled by factors 2, 4, and 8. Results are shown for QCD jets (top rows) and top-quark jets (bottom rows).

well the super-resolution jets estimate the true high-resolution jet information and compare it to how well the low-resolution jets estimate that same quantity. Specifically, for a given observable, we examine the distribution of the quantities:

$$\frac{\text{HR} - \text{LR}}{\text{HR}} \qquad \text{and} \qquad \frac{\text{HR} - \text{SR}}{\text{HR}}. \tag{10}$$

Figure 7 illustrates the relative per-jet performance for LR and SR jets. First, we confirm our observation from Fig. (6) that the variation in the jet mass from down-sampling and from SR up-sampling is small (the distribution peaks near 0). The relative deviation to the HR truth is between 5% and 10%, with an ever so slight bias towards larger jet masses, also visible already in Fig. (6). The increase in the width of the differences for SR is just the effect of our stochastic generative approach for an essentially insensitive high-level observable. We note that the introduction of this extra source of stochasticity may pose challenges for analyses in which the precision of invariant jet mass is central.

For the 2-subjettiness $\tau_2$ the variation is slightly larger than for the jet mass, and both differences have similar widths, so the information loss from down-sampling is almost compensated by the SR up-sampling. Moreover, down-sampling generates a visible bias, while the mean of the difference $\text{HR} - \text{SR}$ is nicely centered around zero. Correlating the two deviations we indeed find that in 54% of all jets the SR image is closer to the HR truth than

**Figure 7:** Distribution of the relative difference in a given high-level observable for per-jet top-jet triplets LR, SR, and HR. In the upper panels we show the normalized deviations with a fitted double sided gaussian where the standard deviation is given by $\sigma(x) = \sigma_l$ for $x < \mu$ and $\sigma(x) = \sigma_r$ otherwise. While in the lower panels we show ratios of deviations from the LR truth with a fitted sigmoid function. The quoted value $p$ gives the fraction of jet triples for which SR is closer to the HR truth than LR truth is to HR truth, indicating that SR has added relevant information.

the down-sampled image. This pattern is enhanced for the 3-jettiness $\tau_3$, where the effect of down-sampling increases in width and in bias. In contrast, the super-resolution difference $SR - HR$ remains at negligible bias, and the increase in width is under better control. Consequently, in the lower panel we see that SR adds information for 61% of all events already for our relative naive and un-tuned approach.

From Fig. (6) we can hope for a dramatic impact of the super-resolution algorithm for $C_{0.2}$, because in spite of the $p_T$-weighting in the observable, the information loss from down-sampling is big. Indeed, the bias in the difference $HR - LR$ becomes of order one, combined with a misleadingly small width of the distribution. The strong peak around an actually wrong value of $C_{0.2}$ implies that down-sampling has indeed lost a significant amount of information. The difference $HR - SR$ is broader than the difference from down-sampling, but as before the bias is better controlled. In the lower panel we confirm that for this non-trivial correlator the SR up-sampling adds significant information about generic jet features and leads to an improvement for 74% of all jets.

The above analysis serves to illustrate that SR indeed adds generic QCD information which translates into established and relevant physics properties. However, while many of the SR variables show measured improvement over LR performance, the benefits of this increased information must be weighed by the increased stochasticity in other variables (i.e. Invariant Jet Mass). The utility of this method therefore depends heavily on the goals and scope of the particular analysis at hand. Future work should make this benchmarking systematic and investigate whether we can further improve the performance of our SR network on crucial correlators.

## 3.4 Model dependence

The ultimate goal for jet super-resolution is to learn jet structures in general, such that SR images can be used to improve multi-jet analyses. In practice, a network could then be trained on some kind of representative jet sample. In our case, the QCD jets and top-quark jets are extremely different, and we further amplify this effect by training the models on one sample and applying them to the other. This gives an example of a large model dependence and allows us to understand the behavior by comparing with the correctly assigned data sets.

In Fig. 8, we show the results from the network trained on QCD jets, now applied to LR top-quark jets. Interestingly, the network generates all the correct patterns for the ordered top-quark jet constituents, albeit with a slightly reduced precision for instance for the 15th constituent. Similarly, the patches still do not include unwanted visible patterns, but are slightly more noisy.

Finally, in Fig. 9 we show the results from the network trained on top-quark jets, but applied to LR QCD-jets. In a detailed comparison with Fig. 4, we see that the network does not generate the more challenging QCD patterns out of the narrow central pixel set. It starts to fail already for the first and second constituents, and works slightly better for the 7th constituent in the transition region before correctly reproducing the soft noise patterns. In the distributions of high-level observables, the problem is most evident in $\tau_2/\tau_1$. Here the training on the top-quark sample pushes the SR QCD-image towards larger values or higher jet multiplicities. This reflects the broader structure of the training sample with its generally larger values of $\tau_2/\tau_1$.

## 3.5 Network Complexity Reduction

The flexibility of deep networks often comes at a cost of complexity. This complexity, in the form of a large number of layers and nodes, means a large number of parameters must be optimized during training. This hyper-flexibility can lead to undesirable side-effects that ultimately hurt its utility especially when it comes to systematic studies. A network with fewer

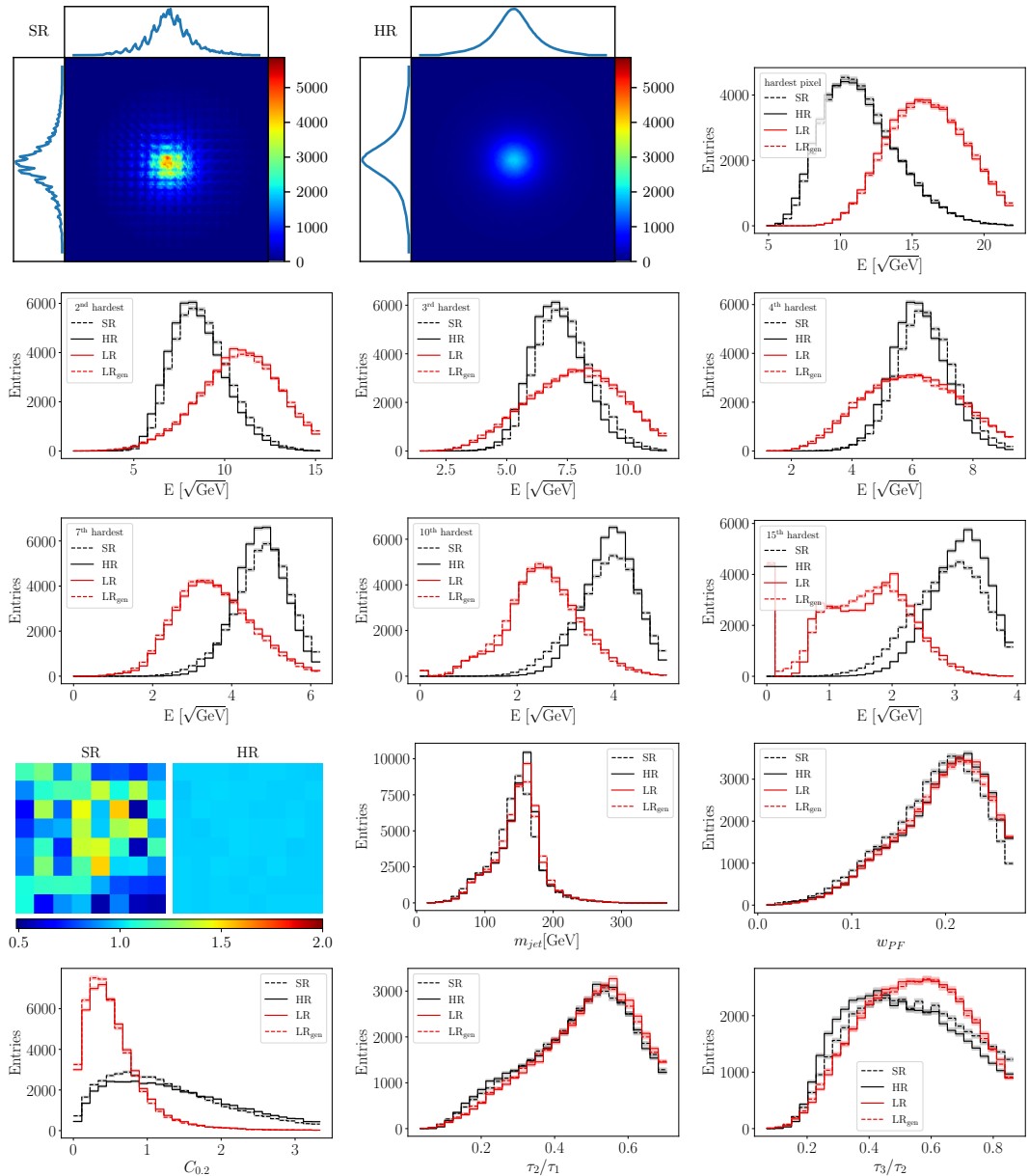

Figure 8: Demonstration of the performance of a network trained on QCD jets and applied to top-quark jets. Top left are averages of the HR and SR images, followed by distributions of the square-root of the energy of leading pixels, sub-leading, etc. Also shown are average $(f \times f)$-patches for the SR and the HR images, and distributions of high-level jet observables, see text for definitions. The zero-bin in energy collects jets with too few entries.

parameters, which achieves the same performance, will be more efficient to train, faster to evaluate, less prone to over-fitting and more likely to generalize. For these reasons, we aim to determine the minimal necessary complexity of our GANs by systematically reducing the number of layers until performance is impacted.

Most of our network complexity resides in the core of the super-resolution GANs, which comprises the residual-in-residual dense blocks (RRDBs), each of which includes 15 convolutional layers. In this section we experiment with a smaller number of blocks, but the same

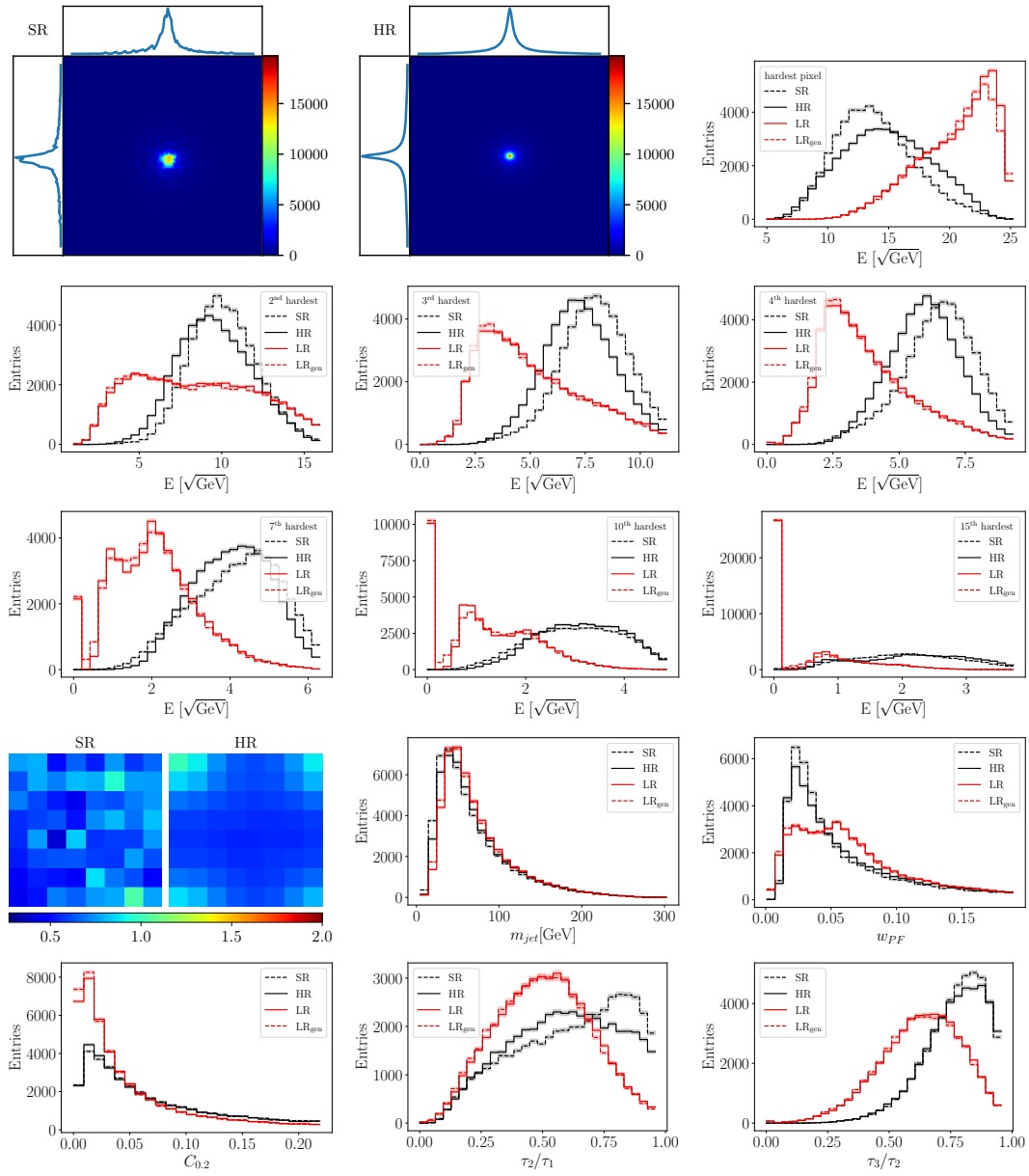

Figure 9: Demonstration of the performance of a network trained on top-quark jets and applied to QCD jets. Top left are averages of the HR and SR images, followed by distributions of the square-root of the energy of leading pixels, sub-leading, etc. Also shown are average $(f \times f)$-patches for the SR and the HR images, and distributions of high-level jet observables, see text for definitions. The zero-bin in energy collects jets with too few entries.

network architecture. In Fig. 10 we compare pixel energy distributions for SR images generated by the reduced-complexity network to those generated by the network described earlier. In the first panels we see that for top-quark jet even a single-block network is able to extract the truth features very well. The remaining challenge is to properly describe the softer pixels, just as we see for the full network in Fig. 5. In the second set of panels in Fig. 10 we show the corresponding result for a network trained on and applied to QCD jets. As expected, the network task is much more challenging because of the smaller number of available LR-pixels

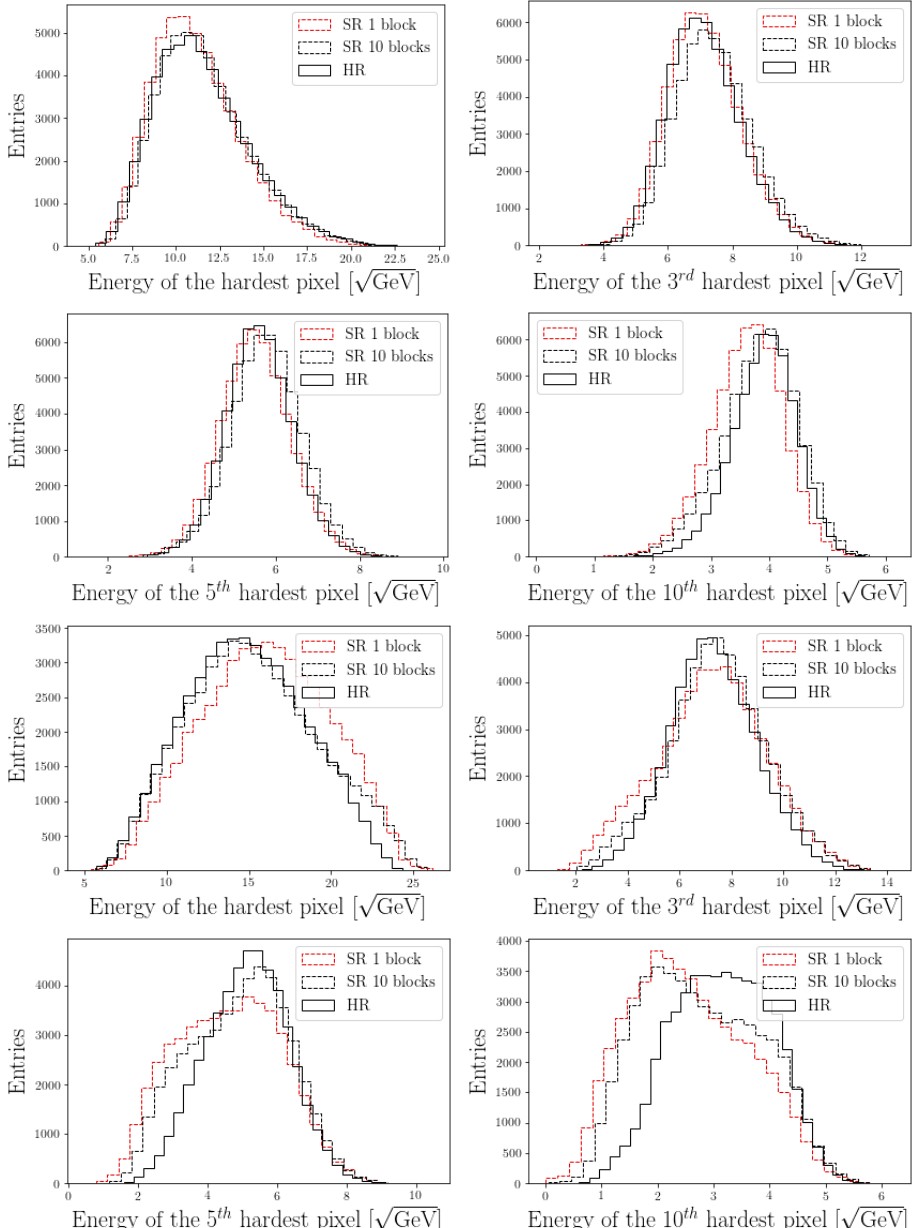

Figure 10: Demonstration of the performance of a reduced complexity (1 RRDB block) network compared to a more complex network (10 RRDB blocks), for networks trained on and applied top-quark jets (upper) and QCD jets (lower). Shown are distributions of the square-root of the pixel energies for the true high resolution image (HR) and super resolution images generated by the reduced and standard complexity network.

and the much more focussed structure of QCD jets. Similar to the full network results shown in Fig. 4, the slim network does not push the energy for the softer pixels to the full truth values, but gets stuck at a slightly softer spectrum.

To illustrate the super-resolution network performance we compute the first Wasserstein distance between the true HR images and the SR images. In Fig. 11 we show this Wasserstein distance as a function of the number of RRDBs for top-quark jets (left) and QCD jets (right). The global scale of Wasserstein distance values reflects the fact that top-quark jets are better

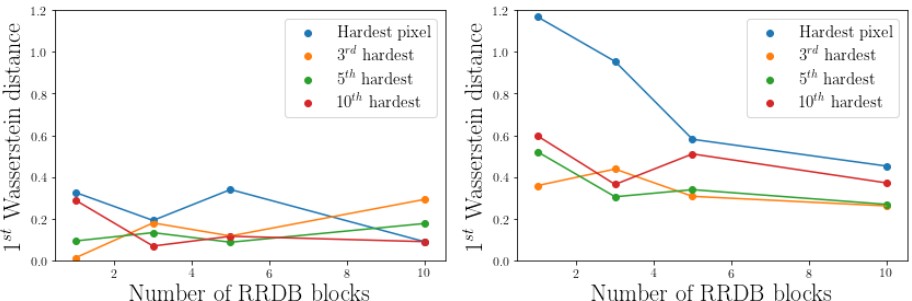

Figure 11: Dependence of the performance of super resolution networks on the number of internal RRDB blocks (See Fig. 2). Performance is measured via the one dimensional Wasserstein distance between the distribution of quantities over true high-resolution images and the super-resolution images. Quantities examined are the energy of the leading pixel, subleading, etc. Left (right) shows results for networks trained on top-quark (QCD) jets and applied to top-quark (QCD) jets.

described by all networks, regardless of the number of RRDBs. As a matter of fact, here the performance improvement from more RRDBs is almost completely covered by the fluctuations from different network initializations and runs. In contrast, the more challenging QCD jets show a significant improvement with an increased network complexity. Interestingly, for both top-quark and QCD jets, the performance improvement is not visibly related to, for instance, hard vs soft pixels. We also emphasize that the larger network complexity required by QCD jets is in contrast to the complexity of the actual jets. While the top-quark jets combine massive decay and QCD splitting patterns, the physics principles behind the QCD jets are much simpler, so the required complexity of the super-resolution network is not driven by the complexity of the underlying objects, but by the effect of the reduced resolution.

## 4 Outlook

Jet physics in terms of low-level observables and with the help of deep networks defines many new opportunities in jet physics and jet measurements at the LHC. For jet classification, or jet tagging, deep networks typically outperform established high-level approaches.

In this paper, we propose a new application of deep learning to jet physics: jet super-resolution, which aims to overcome the limitations of detector resolution and allow for deeper analysis of jet data from ATLAS and CMS. Super-resolution networks can provide additional information, and hence improved resolution, by encoding our knowledge about jet physics in a generative network.

Our results demonstrate that a super-resolution network can indeed reproduce high- resolution jet images of top-quark jets and QCD jets when trained on these samples. We illustrated the performance of the super-resolution networks using images, low-level observables, and high-level observables. The more challenging test of the generality of the network is evaluated by applying a network trained on one sample to jets from the other sample. We confirmed that our super-resolution network exhibits the necessary model independence to be applied to different kinds of jets. This will allow us to train jet super-resolution networks on mixed samples and avoid complications for instance with the poorly defined separation of quark and gluon jets in a QCD sample.

While the main focus of our study was to show that the technique of image super-resolution works reliably on LHC jets, we already showed that it can be used to enhance jet measurements

in regions with poor calorimeter performance. Additionally, we showed that the necessary complexity of the network depends on the source of the jets. Interestingly, equivalent performance on top-quark jets can be achieved with far fewer parameters than QCD jets, despite the former having greater complexities in the underlying physics mechanisms. Such knowledge is helpful in efficiently allocating computational resources when analyzing experimental jet data.

# Acknowledgments

We would like to thank Monica Dunford and Hans-Christian Schultz-Coulon for the experimental encouragement. The research of AB is supported by the Deutsche Forschungsgemeinschaft (DFG, German Research Foundation) under grant 396021762 – TRR 257 *Particle Physics Phenomenology after the Higgs Discovery*. DW is supported by the Department of Energy, Office of Science. JNH acknowledges support by the National Science Foundation under grants DGE-1633631 and DGE-1839285. Any opinions, findings, and conclusions or recommendations expressed in this material are those of the author(s) and do not necessarily reflect the views of the National Science Foundation. All contributions from Julian Collado were done while he was a graduate student at the University of California, Irvine.

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
