# Peer review of "How to GAN Higher Jet Resolution"

_SciPost Physics, doi:SciPost Phys. 13, 064 (2022)_

## Round 1 · Referee Report · Anonymous (Referee 1) · 2021-2-16

Strengths

  1. The paper shows that increasing the granularity of images of hadronic jets based on characteristics learned by a generative neural network recovers the distributions of some jet variables which would be obtained from a high granularity image.

  2. The paper demonstrates some potential benefit of the approach based on comparisons of the distributions of some jet characteristics.

Weaknesses

  1. The model of QCD jets used is not state-of-the-art

  2. The simulation of experimental effects is quite simplistic.

  3. The "resolution" of the images is not translated into resolution of the variables themselves.

  4. The "calorimeter pixel" model of jet images does not reflect the way jets are currently reconstructed in the experiments.

  5. I do not see the claimed model independence demonstrated in the results.

  6. The main conclusion of the paper seems to be already claimed in ref [61] (di Bello et al), with a more realistic (ie particle flow) jet reconstruction.

Report

The use of machine learning to improve the measurement of jets is an interesting area. This paper focusses on improving the "resolution" of images based on pixellated energy, and shows that upscaling the number of pixels to obtain a super-resolution images gives in some cases distributions of jet variables which reflect those obtained from the "true" higher-pixel-number image.

The training samples are a simple simulated QCD jet sample with no extra hard matrix-element emissions, and a simulated sample of jets from top quark decays. Some model independence is claimed in the "Outlook" section but, unless I misunderstand, does not seem to be justified by the results in section 3.3. This is additionally a concern because of the absence of hard emissions in the QCD samples, which one might expect would lead to the high-nsubjettiness tail. (Weaknesses 1, 5 above)

The authors use the word "resolution" to refer to the number of pixels in an image, but the more useful and (in this context) common definition is how well a given variable (e.g. jet mass) is reconstructed. They do not show any resolution metrics in this sense (e.g. jet mass for (HR-SR)/HR would be very interesting) but base their claims on qualitative comparisons of distributions of these variables, which in my opinion is not strong enough evidence to justify the claims made, since in a real experiment one commonly selects on these variables, so getting the distribution right on average is not very interesting unless it is shown that this comes from a genuine improvement in measurement precision (ie resolution in the usual jet measurement sense). (Weakness 3 above)

To turn this into a meaningful experimental resolution, the experimental simulation and reconstruction would need to be much better. For example, the mass resolution is sensitive to the charged/neutral ratio, and matching to tracks via particle flow algorithms, and driven in part by hadronisation; these effects do not seem to be accounted for. Pile up is also absent. These omissions could however be seen as beyond the scope of the paper, which seeks to demonstrate a qualitative effect without quantifying it. (Weakness 2, 4 above)

As it stands, in my opinion the paper is not suitable for publication. If the weaknesses are addressed, especially (3 and 5) this opinion could change, since the idea behind it is definitely interesting. I think it is unlikely to meet the criteria for SciPost Physics however, but could be ok for core.

Requested changes

  1. Please show the resolution on the variables themselves.

  2. Please explain better how the model independence is demonstrated

  3. Please explain how this paper advances from ref [61], or if it is an independent confirmation of some of those results.

  4. Please consider the textual comments in the attached PDF.

  • validity: ok
  • significance: low
  • originality: ok
  • clarity: good
  • formatting: perfect
  • grammar: excellent

Author:  Tilman Plehn  on 2021-07-07  [id 1550]

(in reply to Report 1 on 2021-02-16)

We apologize for taking some time to produce our new results and would like thank the referees for their thoughtful comments, which have improved the draft. Our specific responses to all reports combined are below, we hope that they answer all remaining questions:

%%%%%%%%%%%%%%%%%%%%%%%%%%%%%%%%%%%%%%%%%%%%%%%%%%%%%%%%%% Anonymous Report 3 on 2021-3-8 (Invited Report)

Weaknesses

(1) Model-dependence based on the properties of soft/collinear emissions were not investigated. This is the classic Pythia vs Herwig study. It would be very interesting to find out if Fig 7 is impacted by this.

We thank the reviewer for this suggestion. Such a study would be helpful in establishing the systematic uncertainties of a super-resolution upsampler to be applied to data. Here, however, we have performed an initial study of the viability of up-sampling and its generalization across various topologies, which we think are already interesting. We look forward to such studies in future work.

(2) Imperfect knowledge of the detector is not investigated. Specifically, the resolution of the detector could be systematically biased and the impact on the super-resolved images investigated. Knowing the extent to which the performance of the super-resolved images is affected by imperfect detector knowledge is an important part of the practical application of GANs in this regard.

We agree that application of GANs trained on simulation to real data requires careful studies of the fidelity of the simulation's description of the data. Here, however, we have performed an initial study using simulated samples and a simplified detector description, with no intention to apply this to any real dataset. Such a study would be important for an upsampler applied to data, and so we reserve it for such future studies.

(3) The authors claim in the introduction that data could be used to train the GANs, but I do not see how that is possible. By definition, extrapolating from low-resolution to high-resolution requires a training step for which data does not exist (the high-res image). More details on this would have been useful. Report

The reviewer is correct that high-resolution calorimeter data does not exist in collider data. However, in the last paragraph of the introduction, we describe potential future applications of SR-GANs, including using information from the tracker to provide higher-resolution data which might allow for such training. We have clarified in the text.

This paper attempts to use machine-learning algorithms to improve the resolution of jets that are measured in particle physics collider experiments. The basic approach is to use generative networks to learn the underlying structure of jets, thus improving the resolution of jet images. An appropriate methodology is adopted and the results are presented in detail.

Requested changes

I suggest that the listed weaknesses can be addressed by adding discussion and/or new studies to the manuscript. In order of importance:

(1) Investigate and/or discuss the impact of imperfect detector knowledge. In principle, the pixels in the low-resolution image could be independently smeared by Gaussian to mimic larger-than-default resolution effects. Then the default trained GAN could be applied and the change in the super-resolved image investigated. This would add confidence that the learned behaviour is not too dependent on detector biases.

Thank you for this suggestion. We agree it would be useful, but as per the discussion above, we find it to be out of the scope of this paper, and leave it for future work.

(2) Add a discussion on how data can be used to train the GAN, i.e. if the low-resolution image reflects the detector granularity, then where is the high-resolution image obtained from?

We have clarified the discussion, see above. The HR data is extracted from the tracker, whose resolution is superior to the calorimeter.

(3) Add a discussion/study on the impact of different models for parton showering in jet algorithms. If time permits, the additional study could be to vary the tune of Pythia or to use a different event generator.

Thank you for this suggestion. We agree it would be useful, but as per the discussion above, we find it to be out of the scope of this paper, and leave it for future work.

%%%%%%%%%%%%%%%%%%%%%%%%%%%%%%%%%%%%%%%%%%%%%%%%%%%%%%%% Anonymous Report 2 on 2021-3-2 (Invited Report)

Weaknesses

1- The data set considered in this article is very specific (pythia generated jet images of QCD and top jets for one set of cuts), such that it is difficult to assess the generalisation of the trained super-resolution generative network to other data points.

We have done a first study of model dependence (QCD vs top), but do not claim generalization of the network beyond this.

2- This work follows an earlier study, Ref [61], which has similar conclusions.

A major difference between [61] and our study is that we examine full jets, where [61] studied only super resolution on individual particles. We have added some words to make this clearer in the introduction.

Report

This paper provides an interesting study to improve jet measurements using super-resolution generative networks, showing notably that such methods could help mitigate detector resolution effects. The loss function introduced in section 2, particularly for the generator, is quite complex with multiple separate pieces that are calculated from the generated image data. I would suggest adding some studies of the different pieces and how they impact the trained model, as it is hard to understand from the current study why this specific form of the loss. Does the model perform noticeably worse without the patch loss term? Additionally, it is not clear to me what the sum over std, pow in equation (6) corresponds to, could the authors expand on this? Does this refer to the loss being calculated both with and without reweighting of the pixels to a power p=0.3? Finally, without a detailed hyperparameter scan I do not think the non-optimal parameters in table 1 provide much value to the reader. In section 3, the authors apply their framework to QCD and top jet samples, and compare several observables. It is however hard to read the performance of the model, notably in in the first two plots of the average HR and SR images in figures 4&5, a log of the ratio would be more insightful. Requested changes

1- A discussion of the results of this paper in the context of the previous study in Ref [61].

A major difference between [61] and our study is that we examine full jets, where [61] studied only super resolution on individual particles. We have added some words to make this clearer in the introduction.

2- A more detailed study of the loss function and the impact of its components on the model accuracy.

The patch loss term is strictly not needed, but in the optimization of the hyper-parameters it turned out useful. We added a short discussion to the paper.

%%%%%%%%%%%%%%%%%%%%%%%%%%%%%%%%%%%%%%%%%%%%%%%%%%%%%%%% Anonymous Report 1 on 2021-2-16 (Invited Report)

  1. The model of QCD jets used is not state-of-the-art

We agree that the data set is not new and could be improved. We have made it clear in the text that our data set is the standard top-tagging benchmark data. It is publicly available and well-studied and we believe that these advantages outweigh the disadvantages of being two years old.

  1. The simulation of experimental effects is quite simplistic.

We agree that the simulation is fairly simplistic, but there is a rich tradition in this field of exploring new ideas in a simplified setting. The critical issue is not whether the simulation is the most accurate currently available, but whether it is accurate enough. In this case, it captures the essential nature of the limited-resolution of the jets in a way that allows for a useful study of this new technique. We do not claim that the specific results can be immediately applied to fully realistic simulation or actual collision data.

  1. The "resolution" of the images is not translated into resolution of the variables themselves.

We take this point and have added a new study examining this in detail.

  1. The "calorimeter pixel" model of jet images does not reflect the way jets are currently reconstructed in the experiments.

See response to #2; in the same spirit, we see no reason why jet clustering applied to pixels is a poor model for studying this technique.

  1. I do not see the claimed model independence demonstrated in the results.

Fig 8 and Fig 9 in the revised draft demonstrate that a GAN trained on QCD jets does a reasonable job of upsampling top jets, and vice-versa. The claim is qualitative, not quantitative, which is appropriate for this simplified setting.

  1. The main conclusion of the paper seems to be already claimed in ref [61] (di Bello et al), with a more realistic (ie particle flow) jet reconstruction.

A major difference between [61] and our study is that we examine full jets, where [61] studied only super resolution on individual particles. We have added some words to make this clearer in the introduction.

Report

The use of machine learning to improve the measurement of jets is an interesting area. This paper focusses on improving the "resolution" of images based on pixellated energy, and shows that upscaling the number of pixels to obtain a super-resolution images gives in some cases distributions of jet variables which reflect those obtained from the "true" higher-pixel-number image.

The training samples are a simple simulated QCD jet sample with no extra hard matrix-element emissions, and a simulated sample of jets from top quark decays. Some model independence is claimed in the "Outlook" section but, unless I misunderstand, does not seem to be justified by the results in section 3.3. This is additionally a concern because of the absence of hard emissions in the QCD samples, which one might expect would lead to the high-nsubjettiness tail. (Weaknesses 1, 5 above)

The authors use the word "resolution" to refer to the number of pixels in an image, but the more useful and (in this context) common definition is how well a given variable (e.g. jet mass) is reconstructed. They do not show any resolution metrics in this sense (e.g. jet mass for (HR-SR)/HR would be very interesting) but base their claims on qualitative comparisons of distributions of these variables, which in my opinion is not strong enough evidence to justify the claims made, since in a real experiment one commonly selects on these variables, so getting the distribution right on average is not very interesting unless it is shown that this comes from a genuine improvement in measurement precision (ie resolution in the usual jet measurement sense). (Weakness 3 above)

To turn this into a meaningful experimental resolution, the experimental simulation and reconstruction would need to be much better. For example, the mass resolution is sensitive to the charged/neutral ratio, and matching to tracks via particle flow algorithms, and driven in part by hadronisation; these effects do not seem to be accounted for. Pile up is also absent. These omissions could however be seen as beyond the scope of the paper, which seeks to demonstrate a qualitative effect without quantifying it. (Weakness 2, 4 above)

As it stands, in my opinion the paper is not suitable for publication. If the weaknesses are addressed, especially (3 and 5) this opinion could change, since the idea behind it is definitely interesting. I think it is unlikely to meet the criteria for SciPost Physics however, but could be ok for core. Requested changes

  1. Please show the resolution on the variables themselves.

We thank the reviewer for this insightful comment. We have added a section analyzing the added information per jet by examining the resolution of each variable, as suggested.

  1. Please explain better how the model independence is demonstrated

Fig 8 and Fig 9 in the revised draft demonstrate that a GAN trained on QCD jets does a quite reasonable job of upsampling top jets, and vice-versa. The claim is qualitative, not quantitative, which is very much appropriate for this simplified setting.

  1. Please explain how this paper advances from ref [61], or if it is an independent confirmation of some of those results.

A major difference between [61] and our study is that we examine full jets, where [61] studied only super resolution on individual particles. With the new plots we also offer a comprehensive discussion of the model dependence for our specific top-tagging task, which should be interesting to a broad phenomenology audience. We have added some words to make this clear in the introduction.

Attachment:

2012.11944.pdf

Jonathan Butterworth  on 2021-07-20  [id 1585]

(in reply to Tilman Plehn on 2021-07-07 [id 1550])

I thank the authors for this response and for the additional studies carried out.

I accept the limitations on the scope of the paper regarding the QCD and detector modelling.

The resolution studies (Fig.7) are indeed interesting and in my opinion strengthen the paper. A couple of things I would like to check:

  • I couldn't see sigma_l and sigma_r defined anywhere (sorry if I missed them)
  • when you say the fraction for which "SR is closer to the HR truth than LR truth" this is a bit ambiguous. I think from context you mean SR is closer to HR than LR is to HR, but it could also be interpreted as SR is closer to HR than SR is to LR. Please disambiguate.
  • for many experimental purposes, the introduction of an additional stochastic term (e.g. in fig.7 mass distribution - and it is not that slight, it is nearly a factor of 2!) is a potentially serious flaw in the technique. It would have to be traded off against other advantages such as the reduction in bias seen in C0,2 and nsubjettiness, and the slight improvement in net resolution in nsubjetiness. I think it would help the paper if this were discussed a bit more frankly.

The discussion of the extent to which the technique is model dependent is improved, and I have no further comments there. Similarly, the clarification regarding ref [61] is fine.

Anonymous on 2021-12-07  [id 2014]

(in reply to Jonathan Butterworth on 2021-07-20 [id 1585])

Hey Jon, thank you very much for all the useful comments!

I couldn't see sigma_l and sigma_r defined anywhere (sorry if I missed them)
-> Sorry, now added to the caption of Fig.7

When you say the fraction for which "SR is closer to the HR truth than LR truth" this is a bit ambiguous. I think from context you mean SR is closer to HR than LR is to HR, but it could also be interpreted as SR is closer to HR than SR is to LR. Please disambiguate.
-> Thanks, now clarified in the same caption.

For many experimental purposes, the introduction of an additional stochastic term (e.g. in fig.7 mass distribution - and it is not that slight, it is nearly a factor of 2!) is a potentially serious flaw in the technique. It would have to be traded off against other advantages such as the reduction in bias seen in C0,2 and nsubjettiness, and the slight improvement in net resolution in nsubjetiness. I think it would help the paper if this were discussed a bit more frankly.
-> We appreciate this warning and added a new paragraph to the end of Sec.3.3.

Attachment:

2012.11944.pdf

---

## Round 1 · Referee Report · Anonymous (Referee 2) · 2021-3-2

Strengths

The authors introduce a novel method for up-sampling jet images from low resolution samples, which can be of potential interest in mitigating detector resolution effects. The paper is clear and well written and a useful addition to the literature.

Weaknesses

1- The data set considered in this article is very specific (pythia generated jet images of QCD and top jets for one set of cuts), such that it is difficult to assess the generalisation of the trained super-resolution generative network to other data points. 2- This work follows an earlier study, Ref [61], which has similar conclusions.

Report

This paper provides an interesting study to improve jet measurements using super-resolution generative networks, showing notably that such methods could help mitigate detector resolution effects.
The loss function introduced in section 2, particularly for the generator, is quite complex with multiple separate pieces that are calculated from the generated image data. I would suggest adding some studies of the different pieces and how they impact the trained model, as it is hard to understand from the current study why this specific form of the loss. Does the model perform noticeably worse without the patch loss term? Additionally, it is not clear to me what the sum over std, pow in equation (6) corresponds to, could the authors expand on this? Does this refer to the loss being calculated both with and without reweighting of the pixels to a power p=0.3? Finally, without a detailed hyperparameter scan I do not think the non-optimal parameters in table 1 provide much value to the reader.
In section 3, the authors apply their framework to QCD and top jet samples, and compare several observables. It is however hard to read the performance of the model, notably in in the first two plots of the average HR and SR images in figures 4&5, a log of the ratio would be more insightful.

Requested changes

1- A discussion of the results of this paper in the context of the previous study in Ref [61]. 2- A more detailed study of the loss function and the impact of its components on the model accuracy.

---

## Round 1 · Referee Report · Anonymous (Referee 3) · 2021-3-8

Strengths

(1) The setup of the generative networks is robust

(2) The networks are trained and tested on samples of (i) quark/gluon jets and (ii) jets containing the hadronic decay products of top-quarks. Technical closure is presented for the super-resolved images of jets obtained by applying the trained GAN to the sample it was trained on (i.e. super-resolved images of quark/gluon jets obtained using the GAN trained on samples of quark gluon jets). In addition, model-dependence is tested by applying each trained GAN to the other sample (i.e. super-resolved images of quark/gluon jets obtained using the GAN trained on samples of top-quark jets).

(3) Results are presented in full. The performance of the GAN is good though not perfect. The limitations are clear from the manuscript.

(4) Figure 7 is very interesting: super-resolved images of top-quark jets can be obtained from GANs trained on quark/gluon jets. This implies that once the GAN has learnt the soft/collinear structure of QCD then it can be applied to more complicated objects with jet substructure.

Weaknesses

(1) Model-dependence based on the properties of soft/collinear emissions were not investigated. This is the classic Pythia vs Herwig study. It would be very interesting to find out if Fig 7 is impacted by this.

(2) Imperfect knowledge of the detector is not investigated. Specifically, the resolution of the detector could be systematically biased and the impact on the super-resolved images investigated. Knowing the extent to which the performance of the super-resolved images is affected by imperfect detector knowledge is an important part of the practical application of GANs in this regard.

(3) The authors claim in the introduction that data could be used to train the GANs, but I do not see how that is possible. By definition, extrapolating from low-resolution to high-resolution requires a training step for which data does not exist (the high-res image). More details on this would have been useful.

Report

This paper attempts to use machine-learning algorithms to improve the resolution of jets that are measured in particle physics collider experiments. The basic approach is to use generative networks to learn the underlying structure of jets, thus improving the resolution of jet images. An appropriate methodology is adopted and the results are presented in detail.

The paper satisfies the SciPost conditions for acceptance in the following ways: - Opens a new pathway in an existing research direction, with clear potential for multipronged follow-up work - Provides a synergetic link between different research areas.

Requested changes

I suggest that the listed weaknesses can be addressed by adding discussion and/or new studies to the manuscript. In order of importance:

(1) Investigate and/or discuss the impact of imperfect detector knowledge. In principle, the pixels in the low-resolution image could be independently smeared by Gaussian to mimic larger-than-default resolution effects. Then the default trained GAN could be applied and the change in the super-resolved image investigated. This would add confidence that the learned behaviour is not too dependent on detector biases.

(2) Add a discussion on how data can be used to train the GAN, i.e. if the low-resolution image reflects the detector granularity, then where is the high-resolution image obtained from?

(3) Add a discussion/study on the impact of different models for parton showering in jet algorithms. If time permits, the additional study could be to vary the tune of Pythia or to use a different event generator.

---

## Editorial Decision

published